# Health Professionals’ Perspectives on Commercially Available Intravenous Nutrient Therapies: A Preliminary Report

**DOI:** 10.3390/healthcare12030386

**Published:** 2024-02-02

**Authors:** Monika Karasiewicz, Agnieszka Lipiak, Paulina Jóźwiak, Bogusz Giernaś, Mateusz Cofta, Ewelina Chawłowska

**Affiliations:** 1Laboratory of International Health, Department of Preventive Medicine, Poznan University of Medical Sciences, Święcickiego 6, 60-781 Poznań, Poland; 2Department of Preventive Medicine, Poznan University of Medical Sciences, Święcickiego 6, 60-781 Poznań, Poland

**Keywords:** intravenous nutrient therapies, IVNT, vitamin drips, nutrient cocktails, alternative therapies, health professionals, public health, Poland

## Abstract

Background: Intravenous nutrient therapies (IVNTs) have gained popularity on the commercial market. Targeted at people with a variety of ailments and needs, the procedures allegedly offer numerous benefits and quick results, widely advertised on the websites of drip bars and health clinics as well as in the available literature. What is less often presented is the point of view of the customers of such services and the opinions of health personnel. Although the latter perspective seems to be crucial, little is known about it. Therefore, the purpose of this study was to present the opinions and experiences of health professionals (*n* = 188) on commercially available IVNTs dedicated to adults. Methods: The study was conducted between April 2019 and March 2020 by means of a survey using an ad hoc questionnaire made available mainly to health professionals attending public health postgraduate courses at the Poznan University of Medical Sciences, Poland. Results: As many as 91.5% of the respondents had heard of commercially available IVNTs (mostly from the media), and 47.3% knew of a facility offering such services. Among the possible situations where the use of IVNTs would be justified, the most commonly mentioned was a diagnosed nutrient deficiency (37.8%), while the least common ones were libido problems (1.1%) and the need to speed up metabolism (2.1%). For 25.5% of the respondents, there was no good rationale for using IVNTs. As many as 15.4% had no opinion on the subject. Health risks of IVNTs were recognised by 95.2% of professionals, with the biggest concerns being the lack of full information on patients’ health status and medical contraindications (84%), the risk of overdose and interactions (77.1%), and hypersensitivity or allergic reactions (75.5%). Among the reasons for IVNTs’ popularity, the respondents listed not only fads spread by celebrities and social media (89.4%) and the need for quick, effortless remedies (77.1%), but also reasons inherent in the Polish healthcare system. As many as 80.3% of the respondents stressed the need for public health institutions to take a stand on commercial IVNTs. Knowing of an IVNT facility was not significantly associated with the opinions of professionals in key areas. Conclusion: Postgraduate public health courses are a good opportunity to engage health professionals in discussions about the current challenges, trends, and needs in the area of health promotion and healthcare. This study’s findings shed some light on the opinions about IVNTs held by health professionals, who are important stakeholders of the healthcare system. Thus, these findings may help to better understand the popularity of IVNTs and incorporate health professionals’ perspectives in future efforts aiming to increase the awareness of IVNT-related health risks among both professionals and patients.

## 1. Introduction

Intravenous nutrient infusions administered to patients when justified, i.e., when medically necessary, are an integral part of standard medical care. Recently, however, “on-demand” parenteral administration of vitamins, minerals, and other substances has become a commercially available service at various facilities: drip bars and clinics selling “vitamin drips”, “vitamin cocktails”, or “vitamin injections” [1,2,3,4]. In recent years, the supply of the so-called intravenous nutrient therapy (IVNT) has also developed in Poland [5]. The increased interest in IVNTs has been sparked by marketing practices of drip clinics focused on social media promotion and celebrity endorsement [1,5,6,7,8].

IVNTs are often perceived as a kind of wellness service, and their popularity and market share seem to be on the rise [3,9]. Due to the lack of clarity regarding the effectiveness of IVNTs or clear procedures for their use, they should currently be considered an alternative therapy [2,5,9,10].

On the websites of IVNT facilities, the method is often advertised to patients, as a great remedy “for vitamin and micronutrient deficiencies”, “for weight loss, health recovery, or better physical performance”, “for hangovers”, “for an immunity boost”, “for detox”, “for an anti-ageing effect”, “for a libido boost”, “for healthy skin, hair, and nails” [5]. These websites provide an overview of the offered nutrients. Usually, the mixtures of nutrients that make up a purpose-specific drip have their own marketing names and each is assigned a set of specific promises of beneficial effects [5].

Companies offering intravenous supplementation claim that it has a whole range of health benefits [1,2,7,9,11]. In brief, the arguments used by IVNT proponents are as follows: increased absorption of nutrients, which—thanks to parenteral administration—bypass the digestive system, higher concentrations of nutrients administered parenterally, and quicker effects [10,12]. A rapid overview of the websites offering IVNTs in Poland as well as of the available literature [5] shows that the infusions allegedly lead to increased immunity, prevention of infections, support for other treatments, faster recovery, improved hydration, elimination of fatigue, improved mood, restoration of lost nutrients, improved condition of the skin, hair, and nails, faster muscle regeneration, extra energy, increased physical and mental resistance, and stimulating the body to burn fat.

However, it is worth noting that commercial infusions and injections are poorly studied in terms of both efficacy and safety [5,10,11,12]. To date, there are few publications on their benefits, and all those that exist are anecdotal or lack scientific rigour [2,13]. In some cases, the use of IVNTs has been described as having negative or even fatal effects [1,2,7,14,15]. Notably, many websites of the facilities offering alternative methods, including IVNTs, fail to provide sufficient information on contraindications and do not always specify in detail which medical personnel will administer the procedures [5,13].

There are also few data on customers’ perspectives on commercial IVNTs [2]. A recent survey on the experiences of Poles using such services indicated that customers mainly bought them to increase immunity and stamina or to cope with fatigue, and that they mostly had a good overall experience with the services. However, the study group included only 17 people [2].

An earlier cross-sectional study of the websites of facilities offering parenteral supplementation services in five European countries suggested that the median IVNT experience of such facilities in Poland was below 5 years. This may mean that the IVNT market in Poland is relatively young compared to other European countries, which may indicate that we are currently witnessing a rapid growth [5]. Perhaps this is why the opinion of medical staff has not been presented in the literature yet. Since only physicians, nurses, midwives, and paramedics have the right to administer nutrients parenterally, their opinion on the subject is crucial [5]. We believe that exploring the perspectives of health professionals may complement the available reports presenting the perspective of service users and analyses of the websites of IVNT facilities [2,5]. Given the health risks involved in medically unjustified use of IVNTs, their uncontrollably growing popularity is disadvantageous from the public health perspective, so it is worth taking an interest in this problem and trying to prevent potential threats to patients’ health.

Therefore, the aim of our study was to present the opinions and experiences of medical professionals, mainly physicians, on the functioning of commercially available IVNTs dedicated to adults in Poland. We analysed the respondents’ knowledge of the market of such services and their opinions about the reasons for the growing interest in them among the Polish population. We also asked the participants for opinions about the validity and safety of IVNT use. In addition, we explored the relationships between respondents’ opinions and their gender, profession, holding a degree, knowing of an IVNT facility, and having a patient who used IVNTs.

## 2. Materials and Methods

This study was conducted between April 2019 and March 2020 by means of an anonymous survey. The study protocol was entered into a 2018 competition for young scientists at the Poznan University of Medical Sciences (PUMS). The hypotheses and research methods won the approval of the competition committee of the Faculty of Health Sciences. Since a standardised measure suitable for this study’s aims was not available, a questionnaire was developed in the course of an ad hoc literature review conducted in the Laboratory of International Health of the PUMS Department of Preventive Medicine, whose employees run public health courses for postgraduate health professionals. To develop the questionnaire, the authors used their knowledge and experience of pharmacy, medical analytics, and international public health. In addition, they carried out a prior analysis of the websites of facilities offering commercial IVNTs.

The first part of the questionnaire included demographic questions (gender, age, workplace location, province, profession, and academic degree or title). The next section of the questionnaire, devoted to the respondents’ opinions and knowledge of IVNTs, contained 11 questions, 10 of which were closed-ended. The most common intravenous therapies listed in the questionnaire were determined on the basis of a review of websites of INVT facilities operating in Poland.

The study group included health professionals, mainly physicians attending postgraduate public health courses run by the Poznan University of Medical Sciences, as well as other health personnel studying at the University at that time. Because we used convenience sampling, our study population cannot be considered representative of medical professionals in general. Our respondents were undergoing specialization training and we usually did not know what fields of medicine they represented. All the respondents were informed of the anonymity of the survey and provided informed consent to participate in the study. Under Polish law, this study did not constitute a medical experiment, which was certified in writing by the Chair of the Bioethics Committee of the Poznan University of Medical Sciences. Statistical analyses were performed using PQStat Software (2022) PQStat v.1.8.4.164. The results are presented as absolute values and percentages. A significance level of *p* ≤ 0.05 was used in all analyses. Figure 1 shows the flowchart of the study methodology.

## 3. Results

### 3.1. Study Group

A total of 188 respondents, approached by means of convenience sampling, took part in the study, including 132 women (70.2%) and 56 men (29.8%). This gender bias may have resulted from the fact that in Poland, there are generally more female than male physicians, and this professional group constituted the majority of the sample. The mean age was 31.5 years (SD ± 6.1). The vast majority were from Wielkopolskie province (85.1%) and were working in a city with more than 100,000 inhabitants (78.8%). The sample consisted of 139 physicians (73.9%), 6 dentists (3.2%), 14 dieticians (7.5%), 12 public health graduates (6.4%), 6 paramedics (3.2%), 5 laboratory diagnosticians (2.7%), 1 nurse and 1 physiotherapist (0.5% each), as well as other health professionals (*n* = 4; 2.1%). In total, physicians and dentists made up 77.1% of the study group, while 22.9% were health professionals other than doctors. Degree holders (with a PhD or higher-level degree) constituted 14.4% (*n* = 27).

### 3.2. Familiarity with Commercial IVNTs

As many as 91.5% of the respondents (*n* = 172) answered “yes” to the question “Have you ever heard of commercially available intravenous infusion services offering vitamins or other nutrients?” The remainder had not (*n* = 14; 7.5%) or did not know/did not remember (*n* = 2; 1%). Those who were familiar with the term were also more likely to know of a specific IVNT facility (*p* = 0.000), were female more often than male (*p* = 0.046), and were slightly more likely to be non-physicians (*p* = 0.025). When asked if they knew of any IVNT facility, 47.3% of the respondents (*n* = 89) said yes, 46.3% (*n* = 87) answered no, and 12 respondents (6.4%) said they did not know or remember.

The most frequently indicated sources of knowledge of the subject (multiple-choice question, *n* = 188) were the media (TV, press, radio, the Internet) (*n* = 113; 60.1%), friends, acquaintances, or family (*n* = 83; 44.2%), as well as patients (*n* = 50; 26.6%). Other sources included leaflets, posters in health centres or hospitals (*n* = 44; 23.4%), and doctors (*n* = 35; 18.6%). A total of 16 respondents (8.5%) also indicated display windows of IVNT establishments, advertisements in public spaces, hairdressing and beauty salons, gyms, books, phone calls from representatives of such facilities, or job offers received from them. The least frequently listed sources of knowledge were university (*n* = 14; 7.5%), nurses (*n* = 13; 6.9%), and conferences or symposia (*n* = 9; 4.8%). Only 7.5% of the respondents (*n* = 14) did not cite any source of their knowledge about the commercial offer of intravenous supplementation.

The majority of the respondents (*n* = 137) indicated the IVNT terms they were familiar with. The most popular terms were multivitamin drip (*n* = 101; 73.7%), intravenous supplementation (*n* = 82; 59.9%), vitamin cocktail (*n* = 66; 48.2%), and vitamin bar (*n* = 38; 27.7%). Familiarity with other terms (e.g., hangover drip, detox, vitamin C infusions, L-carnitine) was reported much less frequently (*n* = 8; 5.8%). The name of Myers’ cocktail was known to only 2.9% of the participants (*n* = 4), while 12 respondents (8.8%) had never heard of any of the terms.

The term “intravenous supplementation” was more often recognised by non-physicians (*p* = 0.048), while “vitamin bar” and “vitamin cocktail” were significantly more often recognised by those familiar with an IVNT facility (*p* = 0.002; *p* = 0.008). Familiarity with “Myers’ cocktail” increased with age (*p* = 0.039) and was higher among those with a degree (*p* = 0.041). Those professionals who knew of an IVNT facility were less likely to report an ignorance of any of the terms (*p* = 0.004) (Table 1).

Commercially available intravenous infusions had been used by three of our respondents (1.6%) for the following purposes: to recover, to assess the effectiveness of the procedure, and to support the treatment of pneumonia with vitamin C.

The question “Has a patient ever reported to you the use of commercially available IVNTs?” had 20.2% affirmative responses (*n* = 38) and 60.1% negative ones (*n* = 113), as well as some “not applicable” (10.6%; *n* = 20) and “I don’t know/don’t remember” (9%; *n* = 17) answers. The patients were, according to the participants, “equally often women and men” (*n* = 18; 47.4%) or “mainly women” (*n* = 16; 42.1%). Significantly fewer professionals indicated that it was mostly men who had reported taking such infusions (*n* = 4; 10.5%).

### 3.3. Opinions on Situations in Which the Use of Commercial IVNTs Is Justified

One of the multiple-choice questions listed profiles of people who are typically targeted by IVNT clinics. We presented medical professionals with a selection of the most common offers published on the websites of IVNT clinics and asked them for their opinion. The question was “The following are selected profiles of people who are targeted by offers of commercial intravenous vitamin/nutrient infusions. Please indicate in which situations do you think it is justified to use them (multiple choice possible)”. The professionals could indicate selected offers (consider them justified), respond “I don’t know/difficult to say” or indicate “in none of the above cases”. The most frequently selected options were “patients with diagnosed deficiencies”—37.8% (*n* = 71) and “none of the above”—25.5% (*n* = 48), followed by “persons in need of (alcohol) detoxification”—23.9% (*n* = 45) and “athletes”—23.9% (*n* = 45). The less common choices were overworked/stressed persons (17.0%, *n* = 32) and oncology patients (16.5%, *n* = 31). Fewer than one in ten respondents considered it appropriate to use infusions to improve skin firmness and hydration/for aesthetic reasons (9.6%, *n* = 18), for neurological conditions (9.0%, *n* = 17), or for an immunity boost (8.0%, *n* = 15). The least justified cases were persons with libido problems (1.1%, *n* = 2) and people wishing to speed up metabolism (2.1%, *n* = 4). Other situations, e.g., malabsorption of vitamins and nutrients, sepsis cases, schizophrenia, specialist’s orders, patients after bariatric surgeries, or patients denied treatment within mainstream medicine, were indicated by a total of 7.5% (*n* = 14). As many as 15.4% of the respondents (*n* = 29) did not provide a clear answer to this question, indicating the option “don’t know/difficult to say”.

Male respondents were more likely than women to consider the use of infusions in persons in need of (alcohol) detoxification (*p* = 0.036) and in patients in need of anti-spasmodic treatment (*p* = 0.004). Older participants were more likely to justify IVNTs for the improvement of skin firmness and hydration/for aesthetic reasons (*p* = 0.002), for an immunity boost (*p* = 0.009), and in oncology patients (*p* = 0.012). Knowing of an IVNT facility did not significantly differentiate the responses to this question, but having had a patient who used IVNTs had a greater impact. Respondents whose patients had reported the use of IVNTs were significantly more likely to recognise the validity of their use in overworked/stressed persons (0.042), for increased skin firmness and hydration/for aesthetic reasons (*p* = 0.025), in athletes (intense exercise, recovery) (*p* = 0.019), and in people with libido problems (*p* = 0.014) (Table 2).

### 3.4. Perceptions of IVNTs as Potential Health Risks

The participants (*n* = 188) were also asked if commercial intravenous infusions might pose a risk to patients’ health, and 95.2% of them (*n* = 179) did see such a risk, while 3.7% (*n* = 7) did not know or found it difficult to answer, and a further 1% (*n* = 2) did not perceive the infusions as risky. None of the tested variables (gender, age, profession, holding a degree, knowing of an IVNT facility, having a patient who used IVNT) significantly differentiated the answers to this question. More than 90% in each of the subgroups differing with respect to these variables perceived IVNTs as a risk to patients’ health.

Specific health risks indicated by the respondents were the risks caused by the lack of full information about the patient’s health status and medical contraindications (84%, *n* = 158), the risk of vitamin and mineral overdoses and interactions (77.1%, *n* = 145), and the possibility of hypersensitivity or allergic reactions (75.5%, *n* = 142). As many as 60.6% of the respondents (*n* = 114) considered the use of commercial infusions to be risky due to the lack and/or poor evidence of their efficacy. A risk connected with puncturing the skin (e.g., infection, inflammation) was recognised by 59% of the participants (*n* = 111). Other risks mentioned in the survey included the presence of contaminants or unknown substances in the formulation of the infusions, possible interactions with other drugs, lack of leaflets, lack of certainty about the origin of infusions, lack of clinical trials, and the risk of abandoning conventional therapy in favour of alternative “therapy” 3.7% (*n* = 7).

Women were significantly more likely than men to recognise the risk of vitamin and mineral overdoses and interactions (*p* = 0.006). Physicians and dentists were more likely than other professionals to indicate the lack of full information on the patient’s health status and medical contraindications (*p* = 0.000), the risk of hypersensitivity or allergic reactions (*p* < 0.000001), the lack or poor evidence of IVNT efficacy (*p* = 0.000), and risks associated with puncturing the skin (e.g., infections, inflammation) (*p* = 0.000) (Table 3).

### 3.5. Perceptions of Reasons for the Increased Demand for IVNTs

When asked about the factors influencing the increased interest in commercially available IVNTs in the Polish population, the factor indicated by far the most frequently (89.4%, *n* = 168) was an IVNT trend spread by celebrities, social media, and similar channels. This was the most frequently indicated reason regardless of demographic factors or other variables analysed. The next factor was the need for quick, effortless remedies (77.1%, *n* = 145), which was slightly more often selected by professionals other than doctors (*p* = 0.046). One in three respondents indicated an increasing prevalence of unhealthy behaviours in the population (35.1%, *n* = 66). This factor was significantly more popular with physicians and dentists than with other professionals (*p* = 0.010). Similarly, as many as 32.4% selected aesthetic considerations as a factor that increased interest in commercial infusions (*n* = 61). It was indicated significantly more often by those respondents whose patients reported IVNT use (*p* = 0.023). Less than one in four respondents also indicated “patients’ inability to navigate the Polish healthcare system” (23.4%, *n* = 44). Men were significantly more likely to choose this reason than women (*p* = 0.026). The factors that came next were diagnosed vitamin/nutrient deficiencies (21.8%, *n* = 41) and low accessibility of specialist physicians (17.6%, *n* = 33). Recommendation from a doctor and/or dietician (6.9%, *n* = 13) and an ageing population were the least frequently cited reasons (2.1%, *n* = 4). The last reason was more frequently pointed out by the oldest respondents (*p* = 0.025) and professionals with a PhD or a higher-level degree (*p* = 0.040). In addition, 6.4% of the respondents (*n* = 12) highlighted a number of other factors: lack of trust in doctors, the healthcare system, and recommended therapies; a belief that invasive procedures work better than non-invasive ones; hostile attitudes towards pharmaceutical companies; fear of oncology diseases; ineffectiveness of standard medical therapies (e.g., palliative cancer treatment); a belief in unconventional medicine; alcohol addiction and masking the need for help in combating the addiction; insufficient awareness of potential harm; misinformation from the Internet and forums; lack of knowledge on vitamins/dietary supplements; and lack of access to a dietician within the public healthcare system (Table 4).

### 3.6. The Need for Public Health Institutions to Take a Stand on Commercial IVNTs

As many as 80.3% of the survey participants (*n* = 151) agreed that Polish public health institutions need to take a stand on commercially available intravenous infusions aimed at people without diagnosed nutrient deficiencies. Knowing of an IVNT facility or a patient who used such procedures did not significantly differentiate professionals’ opinions on this topic. Physicians and dentists were more likely to perceive the need for such a statement compared to other professionals (*p* = 0.000) (Table 5).

## 4. Discussion

Intravenous supplementation as a commercial service in Poland appears to be a relatively recent topic and the available research is very limited. From a public health perspective, however, it is difficult to overlook the facilities providing such procedures and the wide range of drips promising effective and readily available help to maintain and improve health. We felt that there was a need for a scientific discussion about the popularity of parenteral supplementation and the validity and safety of IVNT use. It seems that health professionals should have their say here, so we attempted to gauge their opinions. It turned out that nine out of ten professionals had heard of commercially available infusion services and 47.3% knew of a facility that had them on offer. The most frequently cited sources of such knowledge were not only the media but also flyers or posters exhibited in health centres and hospitals (23.4%). Despite this widespread presence of IVNT marketing, universities or conferences were seldom listed as a source of knowledge on intravenous infusions. These findings confirm previous reports which showed that the market for alternative and complementary services is doing well in Poland [5,13,16]. One in five of our respondents had a patient who used IVNTs. This points to quite a wide use of such methods and to a need for publicising and discussing the issue. It is worth mentioning that patients tend to conceal their use of alternative and complementary methods [13,16]. Adequate preparation and openness to discussion could help health professionals more effectively encourage patients to make lifestyle changes and use evidence-based treatments [13]. According to the respondents, nutrient infusions are either used by women and men alike (47.4%) or mostly by women (42.1%), which sheds a little more light on the profile of a Polish IVNT user [2].

To date, no professional Polish language term has been proposed to describe vitamin drips [2]. The available literature offers a few terms which our respondents were familiar with, and the most popular terms were multivitamin drip (73.7%), intravenous supplementation (59.9%), vitamin cocktail (48.2%), and vitamin bar (27.7%). The fact that few professionals had heard of Myers’ cocktail (only 2.9%) and that this knowledge was better among degree holders may indicate that professionals generally do not follow the history or the present development of the IVNT market. Dr John Myers, a physician from Baltimore, Maryland, allegedly pioneered the use of large doses of intravenous vitamins and minerals in combined applications. He used intravenous nutrients in patients with a variety of chronic diseases and is recognised as the inventor of the so-called Myers’ cocktail marketed in the late 1950s [12,17]. According to one Polish study, patients turn to intravenous infusions to boost their immunity, reduce fatigue, improve physical endurance, or get rid of toxins [2]. Among our respondents, 37.8% believed that IVNT use was justified only in patients with diagnosed deficiencies, and a quarter did not see any justification for using such methods. Some professionals (23.9%) were of the opinion that the use of infusions in persons in need of (alcohol) detoxification may be justified, just as in athletes (23.9%). The legitimacy of IVNT use for fatigue or immunity boost was far less recognised. This may confirm earlier findings of a growing scepticism about alternative and complementary methods among Polish doctors, particularly of the younger generations [13]. It seems they hardly ever discuss alternative methods with their patients [16]. At the same time, it was estimated that 2.2% of Polish households used unconventional methods in the pandemic year of 2020 at an average annual cost of PLN 537 (EUR 123) [18].

Among the professionals we surveyed, 16.5% considered the use of IVNTs in oncology patients to be reasonable. Research shows that 24 to even 90% of cancer patients report using at least one complementary or alternative therapy [13,19]. Cancer patients seem to be a prominent target group of facilities selling alternative methods, with up to 46% of them offering treatments with alleged anti-cancer or complementary effect [13]. In Polish complementary and alternative clinics, cancer patients have access to diverse intravenous infusions (vitamin C, alpha-lipoic acid, glutathione, ozone salt, vitamin B complex, and many more) [13]. While it is often difficult to find out who is actually employed in the facilities providing such services, it can be inferred that the staff includes physicians from a variety of medical disciplines [13]. In our survey, the risks to patients’ health from commercially available intravenous infusions were recognised by 95.2% of the respondents. The most frequently indicated risks were the lack of full information on the patient’s health status and medical contraindications, the risk of nutrient overdoses and interactions, and the risk of hypersensitivity or allergic reactions. Compared to the oral route, less is known about the appropriate doses or potential toxic effects of intravenously administered nutrients, and the short- and long-term health effects of such procedures are unclear [20]. There are risks associated with the procedure, such as allergic reactions or infections, especially if the administering staff is not suitably qualified [15,20]. As many as 60.6% of the study participants believed that the use of commercial IVNTs is risky due to the lacking and/or poor documentation of their effectiveness. Almost as many indicated the risk of infection or inflammation. Such concerns seem valid, given previous reports of how such facilities may operate [2,5,15]. In addition, there is not enough research to show that vitamin injections have health benefits or are essential for good health [7,20]. It remains an open question whether and how easy access to parenteral supplementation will affect patient safety, the continuation of mainstream medical therapies, and lifestyle-related health behaviours. Intravenous vitamin intake has become a popular trend in recent years, especially among celebrities [7,20]. Facilities offering IVNTs are active on social media, which are a popular channel for promoting alternative methods [5]. This was noticed by our respondents, who identified the IVNT fad as a key factor increasing interest in commercial intravenous nutrient infusions. Other frequently indicated reasons were the need for quick fixes (77.1%) and the growing prevalence of unhealthy behaviours in the population (35.1%). Intravenous supplementation advertisements often promise spectacular effects and give the false impression that they can help quickly make up for the harm caused by an unhealthy lifestyle. This must be tempting, given the heavy burden of lifestyle-related risk factors in the Polish population, from dietary risks such as low fruit and vegetable consumption and high salt and sugar intake, through high mortality attributable to alcohol consumption, to high prevalence of excessive body weight and lack of recommended levels of physical activity [21].

In our study, we did not pay enough attention to the fact that perhaps patients’ negative experiences with medical staff in Poland and a negative assessment of the overburdened and understaffed public healthcare may be the reason behind the popularity of commercial IVNTs [2,13,21]. Notably, one in four of our respondents felt that patients’ inability to navigate the system may be an important factor. Low accessibility of specialist doctors was also indicated (17.6%). It would be worth taking a closer look at this issue in the future. As many as 80.3% of the survey participants, especially physicians, saw the need for Polish public health institutions to take a stand on commercially available IVNTs dedicated to people without diagnosed deficiencies. Knowing of an IVNT facility or having had a patient who used such procedures did not significantly influence the respondents’ opinions on the subject.

In 2019, the NHS’ medical director warned the public of the potential health risks of using commercial intravenous drips [7]. Two years later, the FDA highlighted concerns about the compounding of drug products by medical facilities under unsanitary conditions [15]. In January 2022, Poland’s Patients’ Ombudsman issued a statement on the violation of collective patient rights related to the use of untested treatments. Having consulted experts in clinical pharmacology and internal medicine, he stated that “administering vitamin infusions without a prior diagnosis of a deficiency of nutrients, vitamins and bioelements contained in the medicinal products prescribed to patients violates patients’ collective rights to quality health services in line with current medical knowledge” [22].

Between 2020 and 2023, the Patients’ Ombudsman issued 36 decisions in which he recognised instances of the use of medicinal products (drugs or vitamin infusions) that were not in line with the current medical knowledge and thus were violating patients’ right to quality health services [23]. These practices may be punishable by a heavy fine. Unfortunately, such decisions are limited in scale and do not gain much publicity. However, coupled with publications such as ours, they may encourage other public health bodies to take a stand on the commercial use of IVNTs. It is crucial for patients to have access to complete information about the procedure if they are to make informed decisions regarding its use.

To the best of our knowledge, our study is the first to analyse the opinions of health professionals on intravenous infusions of vitamins, minerals, and other substances offered commercially in Poland. However, our survey had a number of limitations. We assumed that the results would be preliminary, hence the use of convenience sampling, overrepresentation of women, and the relatively small sample size, which may have biased our conclusions. Furthermore, the fact that participation was not remunerated and, above all, the outbreak of the COVID-19 pandemic, which delegated medical staff to more urgent tasks, resulted in a relatively small number of responses from health professionals. Thirdly, our survey was conducted before the pandemic—an event which may have changed health professionals’ opinions, patients’ behaviours, and IVNT market dynamics. Finally, the responses were collected by means of an ad hoc original questionnaire, which may have provided an incomplete picture of health professionals’ views on IVNTs. Therefore, in our opinion, there is definitely room for future in-depth research that could further explore the factors identified in the present study.

## 5. Conclusions

Our preliminary findings, collected before the COVID-19 pandemic using a convenience sample of medical professionals, are not representative but may suggest the following:Commercial IVNTs are well known to health professionals, as is the related popularity of vitamin drips spread by celebrities and social media. The fact of knowing of an IVNT facility does not influence professionals’ opinions on the indications for using the procedures, their health risks, reasons for their popularity, or the need for an expert opinion on such services;Although IVNTs are fraught with health risks, especially among oncology patients and in persons without diagnosed vitamin/mineral deficiencies, some health professionals justify their use in certain cases. One in four professionals would never justify it, and about 15% have no opinion on the subject. Given that patients use alternative and complementary methods for a variety of reasons, it is not surprising that a high proportion of the respondents are interested in finding out the opinion of public health institutions on intravenous supplementation;Public health courses for health professionals are a good platform for sharing and collecting information about the current challenges, new trends, and needs in the area of health promotion and healthcare. Thanks to this approach, we can present findings which may complement the picture the IVNT market and, ultimately, contribute to a better understanding of IVNTs’ popularity and address it from a public health perspective;There is a need for wider-scale and more representative research among health professionals regarding their opinions on the use of IVNTs. Even in our small sample, however, almost all the respondents saw risks to the patient resulting from the use of commercial nutrient infusions. This should be used as an argument in public discussions about the ways to tackle the popularity of these quack procedures and to limit their availability in Poland. Informing the public about professional opinions on IVNTs, coupled with a social media-based information campaign, might influence both those professionals who take part in the quackery, and the patients who consider its use.

## Figures and Tables

**Figure 1 healthcare-12-00386-f001:**
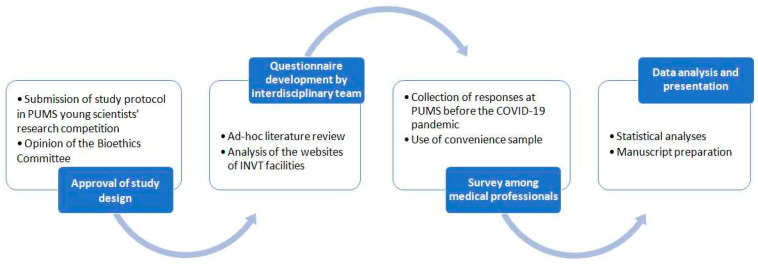
Methodology diagram.

**Table 1 healthcare-12-00386-t001:** Health professionals’ familiarity with commercial IVNTs.

Variable		Study Group		Heard about IVNTs		Knows of a Facility		Knows the Following Terms Related to IVNTs (*n* = 137) ^1^											
		*n*	%	Yes (%)	*p*	Yes (%)	*p*	Multivitamin Drips (%)	*p*	Intravenous Supplementation (%)	*p*	Vitamin Bar (%)	*p*	Myers’ Cocktail (%)	*p*	Vitamin Cocktail (%)	*p*	None of the Terms Listed (%)	*p*
Gender	Female	132	70.2	93.9	0.046 *	43.9	NS *	74.0	NS *	63.0	NS *	28.0	NS *	4.0	NS *	50.0	NS *	6.0	NS *
	Male	56	29.8	85.7		55.4		73.0		51.4		27.0		0.0		43.2		16.2	
	Total	188	100	91.5		47.3		73.7		59.9		27.7		2.9		48.2		8.8	
Age (years) (*n* = 171)	20–29	69	40.4	94.2	NS **	59.4	NS **	77.4	NS **	67.9	NS **	32.1	NS **	0.0	0.039 ***	54.7	NS **	5.7	NS **
	30–39	78	45.6	88.5		39.7		69.2		57.7		23.1		3.8		44.2		11.5	
	40–49	24	14	91.7		41.7		66.7		53.3		40.0		13.3		53.3		6.7	
Profession	MD/dentist	145	77.1	89.7	0.025 *	44.1	NS *	71.3	NS *	54.3	0.048 *	24.5	NS *	1.1	NS *	44.7	NS *	11.7	NS *
	Other	43	22.9	97.7		58.1		79.1		72.1		34.9		7		58.1		2.3	
PhD or higher	Yes	27	14.4	92.6	NS *	48.1	NS *	65.0	NS *	45.0	NS *	35	NS *	10.0	0.042 *	55.0	NS *	20.0	NS *
	No	161	85.6	91.3		47.2		75.2		62.4		26.5		1.7		47.9		6.8	
Knows of a facility	Yes	89	47.3	100	0.000 *	X	X	80.0	NS *	66.2	NS *	38.5	0.002 *	4.6	NS *	61.5	0.008 *	1.5	0.004 *
	No	87	46.3	82.8		X		66.7		52.4		14.3		1.6		38.1		15.9	
	I don’t know	12	6.4	91.7		X		77.8		66.7		44.4		0.0		33.3		11.1	
Had a patient who used it	Yes	38	20.2	97.4	NS *	60.5	NS *	60.5	NS *	50.0	NS *	21.1	NS *	0.0	NS *	36.8	NS *	2.6	NS *
	No	113	60.1	89.4		40.7		48.7		36.3		14.2		0.9		30.1		8.8	

Note: *n*—sample size; *n* = 188 unless otherwise stated; MD—medical doctor; PhD or higher—participant holds a degree of Doctor of Philosophy or a higher-rank academic degree; ^1^ smaller sample size (*n* = 137; 100 female and 37 male participants); * chi-squared test; ** Kruskal–Wallis ANOVA; *** Kruskal–Wallis ANOVA and Dunn–Bonferroni post hoc test; NS—not statistically significant; X—not applicable.

**Table 2 healthcare-12-00386-t002:** Health professionals’ opinions on situations where the use of commercial IVNTs is justified.

Variable		Gender		Age (Years) (*n* = 171)			Profession		PhD or Higher		Knows of a Facility			Had a Patient Who Used It	
		Female	Male	20–29	30–39	40–49	MD/Dentist	Other	Yes	No	Yes	No	I Don’t Know	Yes	No
Study group	*n*	132	56	69	78	24	145	43	27	161	89	87	12	38	113
	%	70.2	29.8	40.4	45.6	14.0	77.1	22.9	14.4	85.6	47.3	46.3	6.4	20.2	60.1
In which situations is the use of commercial IVNTs justified?															
Overwork/stress (%)		15.2	21.4	17.4	17.9	16.7	14.5	25.6	14.8	17.4	18.0	17.2	8.3	26.3	12.4
*p*		NS *		NS **			NS *		NS *		NS *			0.042 *	
To improve skin (%)		9.1	10.7	5.8	6.4	29.2	9.0	11.6	22.2	7.5	12.4	6.9	8.3	18.4	6.2
*p*		NS *		0.002 ***			NS *		0.016 *		NS *			0.025 *	
To speed up metabolism (%)		2.3	1.8	1.4	1.3	4.2	2.1	2.3	3.7	1.9	3.4	1.1	0.0	2.6	1.8
*p*		NS *		NS **			NS *		NS *		NS *			NS *	
In need of detoxification (%)		19.7	33.9	14.5	25.6	37.5	26.9	14.0	37.0	21.7	21.3	28.7	8.3	34.2	21.2
*p*		0.036 *		0.050 **			NS *		NS *		NS *			NS *	
Difficulties concentrating (%)		3.8	3.6	2.9	3.8	4.2	3.4	4.7	0.0	4.3	5.6	2.3	0.0	7.9	3.5
*p*		NS *		NS **			NS *		NS *		NS *			NS *	
In need of immunity boost (%)		7.6	8.9	7.2	5.1	25.0	6.9	11.6	14.8	6.8	10.1	6.9	0.0	10.5	7.1
*p*		NS *		0.009 ***			NS *		NS *		NS *			NS *	
Athletes (%)		21.2	30.4	27.5	23.1	25.0	24.1	23.3	29.6	23.0	20.2	27.6	25.0	39.5	20.4
*p*		NS *		NS **			NS *		NS *		NS *			0.019 *	
Anti-spasmodic treatment (%)		0.8	8.9	4.3	2.6	0.0	1.4	9.3	3.7	3.1	2.2	4.6	0.0	2.6	1.8
*p*		0.004 *		NS **			0.009 *		NS *		NS *			NS *	
Libido problems (%)		0.8	1.8	1.4	1.3	0.0	1.4	0.0	0.0	1.2	1.1	1.1	0.0	5.3	0.0
*p*		NS *		NS **			NS *		NS *		NS *			0.014 *	
Neurological conditions (%)		8.3	10.7	11.6	7.7	8.3	7.6	14.0	7.4	9.3	9.0	9.2	8.3	10.5	7.1
*p*		NS *		NS **			NS *		NS *		NS *			NS *	
Diagnosed deficiencies (%)		37.9	37.5	36.2	42.3	29.2	37.9	37.2	29.6	39.1	31.5	42.5	50.0	47.4	36.3
*p*		NS *		NS **			NS *		NS *		NS *			NS *	
Oncology patients (%)		14.4	21.4	21.7	9.0	33.3	14.5	23.3	18.5	16.1	14.6	18.4	16.7	18.4	14.2
*p*		NS *		0.012 **			NS *		NS *		NS *			NS *	
None of the cases above (%)		26.5	23.2	29	25.6	12.5	24.8	27.9	18.5	26.7	27.0	24.1	25.0	15.8	30.1
*p*		NS *		NS **			NS *		NS *		NS *			NS *	
I don’t know (%)		15.9	14.3	21.7	11.5	8.3	14.5	18.6	11.1	16.1	18.0	11.5	25.0	10.5	13.3
*p*		NS *		NS **			NS *		NS *		NS *			NS *	

Note: *n*—sample size; *n* = 188 unless otherwise stated; MD—medical doctor; PhD or higher—participant holds a degree of Doctor of Philosophy or a higher-rank academic degree; * chi-squared test; ** Kruskal–Wallis ANOVA; *** Kruskal–Wallis ANOVA and Dunn–Bonferroni post hoc test; NS—not statistically significant.

**Table 3 healthcare-12-00386-t003:** Perceptions of IVNTs as potential health risks.

Variable	Study Group			Can Commercial IVNTs Pose a Health Risk to Patients?											
		*n*	%	Yes, They Can Pose a Risk (%)	*p*	Risk of Overdoses and Interactions (%)	*p*	Lack of Full Information on Patient’s Health Status (%)	*p*	Risk of Hypersensitivity or Allergic Reactions (%)	*p*	Risk Connected with Skin Puncture (%)	*p*	Poor Evidence of IVNT Efficacy (%)	*p*
Gender	Female	132	70.2	95.5	NS *	82.6	0.006 *	85.6	NS *	75.0	NS *	59.8	NS *	59.1	NS *
	Male	56	29.8	94.6		64.3		80.4		76.8		57.1		64.3	
	Total	188	100	95.2		77.1		84.0		75.5		59.0		60.6	
Age (years) (*n* = 171)	20–29	69	40.4	94.2	NS **	71.0	NS **	78.3	NS **	68.1	NS **	52.2	NS **	55.1	NS **
	30–39	78	45.6	96.2		82.1		87.2		83.3		60.3		62.8	
	40–49	24	14	100.0		83.3		91.7		79.2		66.7		66.7	
Profession	MD/dentist	145	77.1	96.6	NS *	77.9	NS *	89.7	0.000 *	84.8	<0.000001 *	68.3	0.000 *	69.0	0.000 *
	Other	43	22.9	90.7		74.4		65.1		44.2		27.9		32.6	
PhD or higher	Yes	27	14.4	96.3	NS *	81.5	NS *	96.3	NS *	77.8	NS *	70.4	NS *	66.7	NS *
	No	161	85.6	95.0		76.4		82.0		75.2		57.1		59.6	
Knows of a facility	Yes	89	47.3	96.6	NS *	73.0	NS *	84.3	NS *	76.4	NS *	65.2	NS *	57.3	NS *
	No	87	46.3	93.1		80.5		86.2		75.9		56.3		64.4	
	I don’t know	12	6.4	100.0		83.3		66.7		66.7		33.3		58.3	
Had a patient who used it	Yes	38	20.2	97.4	NS *	78.9	NS *	84.2	NS *	78.9	NS *	63.2	NS *	65.8	NS *
	No	113	60.1	95.6		77.9		86.7		77.9		61.1		60.2	

Note: *n*—sample size; *n* = 188 unless otherwise stated; MD—medical doctor; PhD or higher—participant holds a degree of Doctor of Philosophy or a higher-rank academic degree; * chi-squared test; ** Kruskal–Wallis ANOVA; NS—not statistically significant.

**Table 4 healthcare-12-00386-t004:** Reasons for the increased demand for commercial IVNTs in the Polish population.

Variable		Study Group		Reasons for the Increased Demand for Commercial IVNTs in the Polish Population																	
		*n*	%	Inability to Navigate Polish Healthcare System (%)	*p*	Popularity Spread by Celebrities, Social Media, etc. (%)	*p*	Low Accessibility of Specialist Physicians (%)	*p*	Need for Quick Effortless Remedies (%)	*p*	Diagnosed Vitamin/Nutrient Deficiencies (%)	*p*	Aesthetic Reasons (%)	*p*	Increasing Prevalence of Unhealthy Behaviours (%)	*p*	Doctor’s/Dietician’s Recommendation (%)	*p*	Ageing Population (%)	*p*
Gender (*n* = 188)	Female	132	70.2	18.9	0.026 *	90.2	NS *	16.7	NS *	75.0	NS *	20.5	NS *	32.6	NS *	33.3	NS *	8.3	NS *	2.3	NS *
	Male	56	29.8	33.9		87.5		19.6		82.1		25.0		32.1		39.3		3.6		1.8	
	Total	188	100	23.4		89.4		17.6		77.1		21.8		32.4		35.1		6.9		2.1	
Age (years) (*n* = 171)	20–29	69	40.4	29.0	NS **	89.9	NS **	14.5	NS **	84.1	NS **	20.3	NS **	29.0	NS **	34.8	NS **	7.2	NS **	1.4	0.025 ***
	30–39	78	45.6	21.8		87.2		21.8		74.4		19.2		34.6		39.7		6.4		0.0	
	40–49	24	14	12.5		91.7		20.8		79.2		29.2		41.7		29.2		12.5		8.3	
Profession	MD/dentist	145	77.1	24.1	NS *	89.7	NS *	17.9	NS *	73.8	0.046 *	18.6	0.052 *	31.7	NS *	40.0	0.010 *	3.4	0.001 *	1.4	NS *
	Other	43	22.9	20.9		88,4		16.3		88.4		32.6		34.9		18.6		18.6		4.7	
PhD or higher	Yes	27	14.4	11.1	NS *	852	NS *	18.5	NS *	77.8	NS *	22.2	NS *	40.7	NS *	25.9	NS *	14.8	NS *	7.4	0.040 *
	No	161	85.6	25.5		90.1		17.4		77.0		21.7		31.1		36.6		5.6		1.2	
Knows of a facility	Yes	89	47.3	25.8	NS *	93.3	NS *	16.9	NS *	78.7	NS *	23.6	NS *	28.1	NS *	38.2	NS *	7.9	NS *	1.1	NS *
	No	87	46.3	19.5		85.1		19.5		77.0		23.0		36.8		32.2		5.7		2.3	
	I don’t know	12	6.4	33.3		91.7		8.3		66.7		0.0		33.3		33.3		8.3		8.3	
Had a patient who used it	Yes	38	20.2	18.4	NS *	97.4	NS *	21.1	NS *	73.7	NS *	13.2	NS *	47.4	0.023 *	42.1	NS *	5.3	NS *	5.3	NS *
	No	113	60.1	23.0		86.7		16.8		74.3		24.8		27.4		32.7		7.1		0.9	

Note: *n*—sample size; *n* = 188 unless otherwise stated; MD—medical doctor; PhD or higher—participant holds a degree of Doctor of Philosophy or a higher-rank academic degree; * chi-squared test; ** Kruskal–Wallis ANOVA; ******* Kruskal–Wallis ANOVA and Dunn–Bonferroni post hoc test; NS—not statistically significant.

**Table 5 healthcare-12-00386-t005:** The need for public health institutions to take a statement on commercial IVNTs in persons without diagnosed nutrient deficiencies.

Variable		Study Group		Should Polish Public Health Institutions Take a Stand on Commercial IVNTs in Persons without Diagnosed Nutrient Deficiencies?			
		*n*	%	Yes (%)	No (%)	I Don’t Know /Difficult to Say (%)	*p*
Gender	Female	132	70.2	81.1	7.6	11.4	NS *
	Male	56	29.8	78.6	12.5	8.9	
	Total	188	100	80.3	9.0	10.6	
Age (years) (*n* = 171)	20–29	69	40.4	78.3	7.2	14.5	NS **
	30–39	78	45.6	83.3	11.5	5.1	
	40–49	24	14	91.7	0.0	8.3	
Profession	MD/dentist	145	77.1	84.8	9.7	5.5	0.000 *
	Other	43	22.9	65.1	7	27.9	
PhD or higher	Yes	27	14.4	74.1	7.4	18.5	NS *
	No	161	85.6	81.4	9.3	9.3	
Knows of a facility	Yes	89	47.3	75.3	11.2	13.5	NS *
	No	87	46.3	83.9	8.0	8.0	
	I don’t know	12	6.4	91.7	0.0	8.3	
Had a patient who used it	Yes	38	20.2	81.6	13.2	5.3	NS *
	No	113	60.1	79.6	8.8	11.5	

Note: *n*—sample size; *n* = 188 unless otherwise stated; MD—medical doctor; PhD or higher—participant holds a degree of Doctor of Philosophy or a higher-rank academic degree; * chi-squared test; ** Kruskal–Wallis ANOVA; NS—not statistically significant.

## Data Availability

The data presented in this study are available on request from the corresponding author.

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
