# Peer review of "Health Professionals’ Perspectives on Commercially Available Intravenous Nutrient Therapies: A Preliminary Report"

_healthcare, 2024, doi:10.3390/healthcare12030386_

Round 1
Reviewer 1 Report (Previous Reviewer 1)
Comments and Suggestions for Authors
Overall report of reviewers and authors proofs are evaluated and attached.

Author Response
Please see the attachment

Reviewer 2 Report (Previous Reviewer 2)
Comments and Suggestions for Authors
Revised version of research article entitled “Commercially available intravenous nutrient therapies: Health professionals’ perspectives” is somehow sorted now as per the reviewer’s comments. But still some corrections need to be incorporated in the revised manuscript like:
1. Methodology section could be better in form of flowchart for better understanding of readers.
2. Authors are suggested to incorporate a summarized diagram of whole study after conclusion.
3. Try to simplify the text of whole manuscript by limiting the content in each sections.
4. Add limitations of this study in conclusion section.
5. Also check grammatical errors throughout the manuscript.
Comments on the Quality of English LanguageModerate English editing is required
Author Response
Please see the attachment

Reviewer 3 Report (Previous Reviewer 4)
Comments and Suggestions for Authors
Dear Authors,
Thank you for the revised text and the time you have taken to include the comments made as it has improved your manuscript.
In relation to the number of participants, I would also include within the manuscript the part explanation re the G power assessment as this will add further strength to your discussion and the evaluation of limitations,
Author Response
Please see the attachment

Reviewer 4 Report (Previous Reviewer 3)
Comments and Suggestions for Authors
The article submitted for evaluation concerns an important phenomenon: the existence of commercial methods of intravenous therapy of scientifically unproven utility and safety. The study was conducted using a survey method in people professionally related to medicine. The results are abundant and are presented in quite readable tables. The quality of the results is not impressively high in terms of the scientific values, however the observations and conclusions of the work are practical and concern importand social phenomenon. Quack therapies are, in principle, a bad social phenomenon and should be stigmatized as such. The authors provide constructive criticism of the discussed social problem basing on the questionarre survey study realised on healtcare profesional workers (mainly physicans). The undertaken research problem fits essentialy into the subject of the Journal.
Round 2
Reviewer 2 Report (Previous Reviewer 2)
Comments and Suggestions for Authors
Revised version of research article entitled “Commercially available intravenous nutrient therapies: Health professionals’ perspectives” is now improved as per the reviewer’s comments. It can be considered for publication in its present form.
This manuscript is a resubmission of an earlier submission. The following is a list of the peer review reports and author responses from that submission.
Round 1
Reviewer 1 Report
Comments and Suggestions for Authors
Reviewer report is attached.

Reviewer 2 Report
Comments and Suggestions for Authors
Authors have written a research article entitled “Commercially available intravenous nutrient therapies: Health professionals’ perspectives”. The theme is interesting; however, the manuscript is not properly discussed in context to results and has significant faults related to data/experimental findings of the manuscript that the authors should clarify before a new peer-revision round. The study design of the manuscript is not impressive and does not add some more information related to current research investigations. Therefore, the manuscript is not suitable for publication; it should be carefully revised before the following peer-review process.
• Introduction section is not written in context to address the problem “Intravenous nutrient infusions”. IVNT concept is not properly discussed, which is the main theme of this study.
• In the methodology section, authors used survey-based analysis which is quite not acceptable to current settings of study.
• On the basis of collection of data, authors can not make any conclusion regarding these types of concepts. Further validation in different geographical distributions is needed to confirm these types of studies.
• There is poor presentation of data.
• Discussion is poorly written and divided into subsections which looks like repetition of results again.
• What will be the output of these types of studies in context to addition of information related to IVNT?
Comments on the Quality of English Language
Moderate English changes are required
Reviewer 3 Report
Comments and Suggestions for Authors
The article submitted for evaluation concerns an important phenomenon: the existence of commercial methods of intravenous therapy of scientifically unproven importance. The study was conducted using a survey method among people professionally related to medicine, mainly doctors. The results are presented in tables that are difficult to read and sometimes do not fit on one page. Can these tables really not be reorganized to be more readable and located on one page instead of two? Tables contains text written both vertically and horizontally. It requires quite a bit of vertical and horizontal head gymnastics to read the text on a desktop computer. Only table 5 is legible. The quality of the results is not impressively high in terms of the scientific values. In my opinion, the conclusions of the work should be made more practical. Quack therapies are, in principle, a bad social phenomenon and should be stigmatized as such. The authors are not too critical of this fact and in their conclusions they do not provide a way to limit this phenomenon. In my opinion it could be the most valuable issue of the article.
Reviewer 4 Report
Comments and Suggestions for Authors
Dear Authors,
Thank you for the opportunity to review your manuscript “Commercially available intravenous nutrient therapies: Health professionals’ perspectives” Although interesting, the manuscript gives no overview on why Intravenous nutrient therapies (IVNTs) are of value as they have not been evaluated within the text to state why at all they would be of value for a health professional to understand.
Line 38 you state: “Study findings shed some light on the functioning of the IVNT market and may help to better manage this phenomenon from a public health perspective.” Please expand this as it is not clear what phenomenon you are referring to and why are IVNT’s important over current therapies and what health conditions are they relating to.
Line 86: You state” There are reports suggesting that the IVNT market in Poland is relatively young compared to other European countries, which may mean that we are currently witnessing a rapid growth” You state studies and yet only offer one reference?? Please explain. Also how have you assessed the situation to show there us rapid growth? What does this mean?
Line 103 you state” The questionnaire used was developed in the course of an ad-hoc literature review conducted in the Laboratory of International Health of the Poznan 104 University of Medical Sciences” please explain the reason for this being established at this location, why, how was it evaluated and how as it standardized?
Line 123: you stated use of 188 respondents, why was this number chosen and if a G power was to evaluate this, would it give you a understandable outcome and a correlation that may be acceptable? This is not clear, please explain. With 132 women responder, how have you dealt with gender bias??
Line 133: You state “As many as 91.5% of the respondents (n=172) had heard of commercially available (paid) IVNTs.” Please explain the use of the term :heard” how was this reported and evaluated in context of any treatment plans to be used, also what is paid??
Line 178: You state” One of the multiple-choice questions listed profiles of people who are typically targeted by IVNT clinics. The participants were asked to indicate those in which the use of IVNTs was, according to them, justified. The most frequently selected options were “patients with diagnosed deficiencies” This is not clear to the assessment of deficiencies and where these common amongst the respondents and to what extent in relation to the multiple options offer to the respondents??
Line 194 You state: “Male respondents were more likely than women to consider the use of infusions in persons in need of (alcohol) detoxification……………………..Please explain why this is the case as it is not clear.
Line 209: You state” The participants (n=188) were also asked if commercial intravenous infusions might pose a risk to patients’ health,” Please explain how this was evaluated and what health risks are you referring to??
Please define what you consider “Risk” to be as it is the foundation of discussion.
Line 227: You state” Women were significantly more likely than men to recognize the risk of vitamin and mineral overdoses and interactions” Please explain what interactions you are referring to, what mineral and vitamins you are referring to and how have they been evaluated as it is not clear.
Line 235: You state” Among the factors influencing the increased interest in commercially available IVNTs in the Polish population, an IVNT fad spread by celebrities, social media and similar channels was indicated by far the most frequently” If it is a FAD, how has this been establishes, what data is there it substantiate this claim and why is this overriding any medial requirements. This is not clear, please explain.
Please note section 4.3 & 4.4 need to be re-evaluated and the term risk overdose, long term effect, allergic reaction ect need substantial explanation.
Line 424: You state” Thanks to this approach, we can present findings which may complement the picture the IVNT market and ultimately contribute to a better management of this phenomenon from a public health perspective.” What management are you referring to as it is stated this is a FAD ect and that it seems to be outside the health professionals sphere of influence. This claim has not be established within the manuscript and what is better management of the phenomenon??
Comments on the Quality of English LanguagePlease note there are some sentences that need to be reviewed as the meanings are not clear.